# A Gated-Recurrent-Unit-Based Interacting Multiple Model Method for Small Bird Tracking on Lidar System

**DOI:** 10.3390/s23187933

**Published:** 2023-09-16

**Authors:** Bing Han, Hongchang Wang, Zhigang Su, Jingtang Hao, Xinyi Zhao, Peng Ge

**Affiliations:** 1Sino-European Institute of Aviation Engineering, The Civil Aviation University of China, Tianjin 300300, China; albanhanbing@hotmail.com (B.H.); mailwhc@163.com (H.W.); jthao_siae@126.com (J.H.); xy-zhao@cauc.edu.cn (X.Z.); 2The 38th Research Institute of China Electronics Technology Group Corporation, Hefei 230093, China; gepeng09@gmail.com

**Keywords:** small bird tracking, target tracking, gated recurrent units, interacting multiple model, Lidar

## Abstract

Lidar presents a promising solution for bird surveillance in airport environments. However, the low observation refresh rate of Lidar poses challenges for tracking bird targets. To address this problem, we propose a gated recurrent unit (GRU)-based interacting multiple model (IMM) approach for tracking bird targets at low sampling frequencies. The proposed method constructs various GRU-based motion models to extract different motion patterns and to give different predictions of target trajectory in place of traditional target moving models and uses an interacting multiple model mechanism to dynamically select the most suitable GRU-based motion model for trajectory prediction and tracking. In order to fuse the GRU-based motion model and IMM, the approximation state transfer matrix method is proposed to transform the prediction of GRU-based network into an explicit state transfer model, which enables the calculation of the models’ probability. The simulation carried out on an open bird trajectory dataset proves that our method outperforms classical tracking methods at low refresh rates with at least 26% improvement in tracking error. The results show that the proposed method is effective for tracking small bird targets based on Lidar systems, as well as for other low-refresh-rate tracking systems.

## 1. Introduction

Bird strikes are a kind of collision between a bird and an aircraft in a moving state. With the rapid development of civil aviation, the number of aircraft flying in the air has increased continuously and rapidly, leading to an increasing number of bird strike events [1]. Direct and indirect losses due to bird strikes are estimated to be over USD 124 million in 2021 alone according to the report of the Federal Aviation Administration [2,3]. Bird strikes pose a significant challenge to aviation safety, and bird monitoring and early warning in airports and their surrounding airspace is of great significance to ensure the safety of civil aviation operations. However, birds are typically low-flying, small, and slow-moving targets that are difficult to track effectively with conventional microwave radar [1,4]. Single-photon counting Lidar is a kind of Lidar using time-correlated single-photon counting (TCSPC) detection technology [5], which not only inherits the high-resolution and high-accuracy detection capability of Lidar but also greatly improves the detection sensitivity of the system to the target echo photons thanks to the use of TCSPC technology. Therefore, single-photon counting Lidar has emerged as a promising solution for the detection and tracking of bird targets at airports.

However, the use of TCSPC technology in Lidar requires more time to scan the target in order to accumulate sufficient echo photons and thus to reduce the impact of background light noise, resulting in a low scanning frequencies. This limitation leads to fewer sampling points for the target under surveillance and makes it difficult to detect changes in target motion; bird trajectories thus exhibit characteristics similar to manoeuvring targets with multiple motions and high mobility. As shown in Figure 1, we can see the target motion change is clear and easy to track when the sampling frequency is high, but the target motion change is difficult to track at low sampling frequencies. Furthermore, the characteristics of the motion transition process also vary over the course of the target’s moving, which increases the complexity of bird tracking. Therefore, the first problem to be solved is the inability to represent the motion model due to the low sampling frequency. For small bird tracking in a Lidar system, the small bird is only a single point on the Lidar scanning map, which is indistinguishable in terms of photon echo intensity and size from false-alarm points of Lidar due to background noise. Since the changes in the target motion are difficult to predict precisely, false-alarm points have the probability to be considered the target with motion changes; this fact also increase the difficulty of small bird tracking. Therefore, the second problem to be solved is the false-alarm point filtering. To enable Lidar in bird surveillance in airports and their surrounding airspace, this paper focuses on small bird tracking on a Lidar system under the low-sampling-frequency condition to develop a nonlinear bird model representation method and a false-alarm point filtering method.

For the tracking problem, classical tracking methods, including Kalman filter (KF) [6,7], extended Kalman filter (EKF) [8,9], unscented Kalman filter (UKF) [10,11], and particle filter (PF) [12,13], are commonly used for target tracking. The KF method is an efficient recursive state estimation method which can give optimal state estimation when the target can be described by a linear model and thus has a good tracking performance in long-distance bird migration [14]. In [6], the authors combined the KF method with belief propagation to achieve manoeuvring multi-target tracking, but it is not suitable for regional area bird tracking. Based on the KF method, the EKF method linearlizes the nonlinear model at each estimated state to propagate an approximation. Although the EKF method is widely used in practice, the EKF method is only able to linearize the given nonlinear model and does not allow for model modification, making it difficult to deal with motion changes during bird target tracking in a Lidar system. The UKF and PF methods use a probability model in place of a determined motion model, and they show good tracking capability for nonlinear manoeuvring targets. However, limited by the probability update method that usually requires multiple samples for an accurate estimation, they may not be able to effectively track bird trajectories with multiple motions at low sampling frequencies due to the lack of sampling for one motion. All these methods are based on a single dynamical model, which cannot achieve satisfactory performance when the motions of manoeuvring targets show great complexity and diversity and vary unexpectedly [15,16]. To track high manoeuvring targets, the interacting multiple model (IMM) algorithm has been proposed [17,18,19], where multiple models are used in parallel to track the target and their estimates are weighted and combined to give a final estimate and prediction. In [20], the authors combined the IMM mechanism and PF method to track manoeuvring targets, and a new modified IMM based on UKF was proposed to track ballistic missile motion in [21]. All these approaches provide interesting tracking performances. However, the performance of the IMM method still depends on the choice of determined motion models, and the IMM mechanism can select or combine the results of multi-models based the models’ probability, but it cannot update the models to keep up with the motion change in the manoeuvring targets. Moreover, the models’ probability in the IMM mechanism is a difficult task to be dedicated with an analysis by leveraging a priori information [21]. And, performance degradation appears when the number of models used in the IMM mechanism to describe the manoeuvring target motion increases [22].

In recent years, deep learning-based target detection and tracking methods have attracted the attention of many scholars. In contrast to classical tracking methods, which rely on mathematical models and assumptions about the target’s motion, deep learning-based methods offer the potential to learn complex motion patterns directly from data. deep neural networks (DNNs) methods, such as long short-term memory (LSTM) [23,24], gated recurrent unit (GRU) [25,26], and Transformer [27,28] have been introduced for handling time series information in target tracking problems and have achieved significant results in pedestrian trajectory tracking and multiple object tracking, thanks to their ability to extract time series features and to generate nonlinear models. In [29], authors proposed a trajectory tracking method using feed-forward neural networks instead of KF methods, which significantly increased the tracking performance, even though the computational complexity was higher than that in traditional methods. In [30], the authors established a bi-directional LSTM-based manoeuvring target tracking algorithm for civil aircraft tracking to solve the problem that manoeuvre motion cannot be modelled in time with predefined multiple models. The neural network can track manoeuvring targets after sufficient training, but the large complexity affects the practical application. The authors in [31,32] combined the theory of DNN and traditional tracking filters, proposing an IMM-LSTM model algorithm to track manoeuvring targets in different scenarios and achieved promising performance. However, this method requires input data at high sampling rates, and a significant degradation appears when the sampling time interval is larger than 1 s, which is not suitable for track trajectories effectively at low sampling frequencies. The combination of DNN and model-based tracking methods has also been explored [33,34,35] at a temporary low-sampling-rate condition; the authors proposed to use the prediction data from the DNN method when a measurement is not available in the system failure condition and thus ensuring that the tracking process of a Kalman filter works properly. These applications also require the help of high-sampling-frequency-sampled historical data, which cannot be satisfied in a Lidar system since it always works at low sampling frequencies.

To better illustrate the state of the art, we classify the tracking methods according to three standards: (1) the quantity of models; (2) the basis of models; and (3) the ability to track a manoeuvring target, as shown in Figure 2. Regarding the quantity, the tracking methods can be separated into single model and multiple models. Regarding the basis, the tracking methods can be categorized as data-driven and model-driven. Regarding the ability to track a manoeuvring target, the methods can be classified as a weak ability model, a medium ability model, and a strong ability model. Different from other existing tracking methods, we focus on data-driven methods with multiple models to achieve a strong ability to track manoeuvring targets, so that we can handle the situation with low sampling frequencies.

In this paper, focusing on the urgent needs to address the current bird tracking problem on Lidar systems, we address two main problems: (1) bird tracking under the low-sampling-frequency condition and (2) false-alarm point filtering. A novel approach named IMM-GRU is proposed by combining a gated recurrent unit (GRU)-based motion model with the interacting multiple model (IMM) mechanism. Inspired by the capability that deep learning methods have shown in model learning based directly on data, we use a GRU-based network to extract the motion features of bird targets. Since the architecture of a GRU network is usually complex and costly, multiple small GRU architectures are implemented to learn different independent motion patterns at different sampling frequencies. In order to select and fuse the most suitable GRU-based motion model for trajectory prediction and tracking, the interacting multiple model mechanism is dynamically integrated and the probability function is built to evaluate each model. Finally, a simple back-tracking correction mechanism is proposed to reduce the impact of a false-alarm point. The experimental results demonstrate the effectiveness of our proposed method in bird trajectory tracking, surpassing the performance of conventional tracking methods. The main technical contributions of this paper are as follows:•The development of an IMM-GRU tracking method that combines the IMM mechanism and GRU-based nonlinear motion models to enable effective tracking of bird targets at low sampling frequencies.•The introduction of an approximation state transfer matrix to approximate the explicit representation of the GRU-based motion model, which enables the fusion of the GRU-based motion model and IMM mechanism, and model selection.•The use of the back-tracking correction mechanism improves the prediction performance and the robustness to false-alarm points.

The structure of this paper is organized as follows. In Section 2, we provide an introduction to our proposed IMM-GRU method for manoeuvring target tracking. We explain how the IMM mechanism is integrated with GRU networks to enable effective tracking of bird trajectories. In Section 3 and Section 4, we present the results of numerical simulations that assess the performance of our proposed algorithm. We compare its performance with several classical methods commonly used in target tracking. In Section 5, we conclude the paper with a summary of our findings and potential future research directions in the field of bird trajectory tracking.

## 2. IMM-GRU Method

As bird targets need to overcome gravity in flight with the help of wind, the movement of bird targets usually changes slowly in the altitude direction, which is easy to track. Based on this motion characteristic, the tracking of bird targets can be divided into two separate parts: horizontal position tracking and altitude position tracking. Only the horizontal position tracking part is affected by a low Lidar sampling frequency, and our bird target tracking problem can therefore be reduced to a two-dimensional planar tracking problem.

The aim of tracking problem that this paper addresses is to estimate the states of a bird target at time *k* and to predict the states at time k+1 according to the sequence of observations measured up to time step *k*. The moving model and measurement model of bird target are described as follows:
(1)$$\begin{array}{ccc} X_{k+1} & = & f\left(X_{k}\right)+q_{k} \end{array}$$
(2)$$\begin{array}{ccc} Z_{k} & = & HX_{k}+r_{k} \end{array}$$
where $X_{k}={\left[x_{k},v_{x_{k}},y_{k},v_{y_{k}}\right]}^{T}$ denotes the target state at time *k* and $Z_{k}={\left[x_{k}^{o},v_{x_{k}}^{o},y_{k}^{o},v_{y_{k}}^{o}\right]}^{T}$ represents the measurement at time *k*. Since Lidar measures the distance of a target by using a laser pulse, the velocity measurement of the target based on the Doppler effect of the microwave echo in a conventional radar is no longer applicable for the laser pulse due to high laser frequencies, so we use the difference between two adjacent observed positions as the observed velocity of the target. H is the noiseless measurement matrix, f(·) is the time-varying nonlinear state transition function that presents the bird’s moving process. $q_{k}$ and $r_{k}$ are the process noise and measurement noise, and ${\left(\cdot \right)}^{T}$ is the transpose operation.

The principle of IMM-GRU is depicted in Figure 3, which includes four steps: noise filtering, state estimation, model probability update, and trajectory prediction. In this approach, at each time step, we first filter out the false-alarm measurements based on the prediction at the previous time step. Afterwards, the estimated state is derived based on the previous prediction and the measurement at the current time step. Subsequently, the probabilities of the motion models are updated based on the measurement and covariance of the estimated state. Finally, *n* motion models are employed to match the change in motion and to predict the target trajectory, and the prediction is combined based on the updated probabilities. The following parts will successively present the process and function of each step in detail. Since all the processes are cycled over time, we start from the trajectory prediction step to make it easier to understand.

### 2.1. Trajectory Prediction

At low sampling frequencies, the motion change information of a bird target is very complex: a target can change motions several times during a sampling time interval, and it can also maintain the same motion for long time. To address the variability in the motion model of small bird targets under the low sampling frequency condition of Lidar, we developed multiple GRU-based deep neural networks for target trajectory prediction based on the previous *L* historical observation data. Each GRU-based model was trained with designed data in order to learn a motion change rule that is different from the others. The proposed strategy tries to give all potential possible predictions.

The GRU-based neural network framework is shown in Figure 4: the GRU network consists of an input layer, a hidden layer, and an output layer. The required historical data input length is *L*. Take the *j*th GRU-based model as an example; in the input layer, the measurements and state estimates of the target at time step *k* are spliced to feed the GRU network as input $Y_{k}^{j}$, which is written as follows:(3)$$\begin{array}{ccc} Y_{k}^{j} & = & \mathrm{concat}({\hat{X}}_{k}^{j},Z_{k}) \end{array}$$
where ${\hat{X}}_{k}^{j}$ is the estimated state given by the *j*th model. The simultaneous input of measurement and estimation information into the network helps the network to extract the information embedded in measurement and estimation. This strategy can not only reduce the prediction error caused by the mistake of treating a false-alarm point as a measurement with the help of the measurements but also adaptively retains information about the motion changes embedded in the measurement based on the estimation information.

The hidden layer consists of *M* bidirectional GRUs which are stacked together, with one bidirectional GRU taking in outputs of the previous bidirectional GRU, computing its output, and giving the output to the next bidirectional GRU. The GRUs extract successively useful information from the input and then update the features stored in the hidden layer, which are extracted from the historical measurement $Z_{0:k}$ and states ${\hat{X}}_{0:k}^{j}$. In the GRU-based network, the processing of the *m*th bidirectional GRU at time step *k* can be described as follows:(4)$$\begin{array}{cc} {\vec{h}}_{k}^{\left(m\right)} & =\mathrm{GRU}(h_{k}^{\left(m-1\right)},{\vec{h}}_{k-1}^{\left(m\right)}) \end{array}$$(5)$$\begin{array}{cc} {\overset{\leftarrow }{h}}_{k}^{\left(m\right)} & =\mathrm{GRU}(h_{k}^{\left(m-1\right)},{\overset{\leftarrow }{h}}_{k-1}^{\left(m\right)}) \end{array}$$(6)$$\begin{array}{cc} h_{k}^{\left(m\right)} & ={\vec{h}}_{k}^{\left(m\right)}⊕{\overset{\leftarrow }{h}}_{k}^{\left(m\right)} \end{array}$$
where ${\vec{h}}_{k-1}^{\left(m\right)}$ is the forward hidden feature, ${\overset{\leftarrow }{h}}_{k-1}^{\left(m\right)}$ is the backward hidden feature, $h_{k}^{\left(m\right)}$ is the output of the *m*th hidden layer, ⊕ is the element-wise sum operation, and GRU(·) is the GRU operation. The initial hidden feature $h_{k}^{\left(0\right)}$ at time step *k* is the input of GRU network $Y_{k}^{j}$.

The final features $h_{k}^{\left(M\right)}$ of the *M*th bidirectional GRU are given to the output layer, which consists of a full connection and tanh activation functions. Finally, the network gives the prediction $X_{k+1/k}^{j}$.
(7)$$\begin{array}{ccc} X_{k+1/k}^{j} & = & \tanh(W_{h}h_{k}^{\left(M\right)}+b_{h}) \end{array}$$
where $W_{h}$ is the weight matrix of the final hidden state and $b_{h}$ is the bias coefficients. Then, the prediction $X_{k+1/k}$ is given by combining all predictions of *n* motion models:(8)$$\begin{array}{ccc} X_{k+1/k} & = & \sum_{j=1}^{n}\mu_{k}^{j}X_{k+1/k}^{j} \end{array}$$
where $\mu_{k}^{j}$ is the probability of the *j*th motion model at time *k*, and the calculation of $\mu_{k}^{j}$ will be described in the probability update process.

### 2.2. Noise Filtering

After the trajectory prediction, *n* different predictions are given by *n* different motion models. In this step, we first find the approximate transfer matrix to describe the transition from the current estimated state to the prediction. Then, we give the range of possible locations of the target at time k+1 and the corresponding probabilities. Finally, we filter the false-alarm points based on the probabilities given in the previous step.

Since the GRU network can neither output a representation of the displayed state transfer matrix nor the probabilities of a range, the prediction of the *j*th motion model $X_{k+1/k}^{j}$ given by the GRU-based network is only a prediction point, which does not offer enough range information to filter noise. Therefore, we first construct an approximate one-step state transfer matrix based on a combination of the constant velocity (CV) model and constant turn rate (CT) model:(9)$$\begin{array}{ccc} F_{k+1/k} & = & \alpha F_{\mathrm{CV}}+\left(1-\alpha \right)F_{\mathrm{CT}} \end{array}$$(10)$$\begin{array}{ccc} \alpha & = & \arg\min_{\alpha \in [0,1]}{∥F{\hat{X}}_{k}^{j}-X_{k+1/k}^{j}∥}_{2} \end{array}$$
where ${\hat{X}}_{k}^{j}$ is the estimated state of the *j*th motion model at time *k*; α is a coefficient of the model; and $F_{\mathrm{CV}}$ and $F_{\mathrm{CT}}$ are, respectively, the state transfer function of CV model and CT model, which are expressed as follows:(11)$$\begin{array}{ccc} F_{\mathrm{CV}} & = & \left[\begin{array}{cccc} 1 & T & 0 & 0\\ 0 & 1 & 0 & 0\\ 0 & 0 & 1 & T\\ 0 & 0 & 0 & 1 \end{array}\right] \end{array}$$(12)$$\begin{array}{ccc} F_{\mathrm{CT}} & = & \left[\begin{array}{cccc} 1 & \frac{\sin(\Omega T)}{\Omega } & 0 & -\frac{1-\cos(\Omega T)}{\Omega }\\ 0 & \cos(\Omega T) & 0 & -\sin(\Omega T)\\ 0 & \frac{1-\cos(\Omega T)}{\Omega } & 1 & \frac{\sin(\Omega T)}{\Omega }\\ 0 & \sin(\Omega T) & 0 & \cos(\Omega T) \end{array}\right] \end{array}$$
where Ω is the angular velocity in the CT model and *T* is the time interval between two adjacent steps.

Afterward, we calculate the covariance of prediction $S_{k+1/k}$ based on the approximate one-step state transfer matrix $F_{k+1/k}$:(13)$$\begin{array}{ccc} S_{k+1/k} & = & F_{k+1/k}S_{k}F_{k+1/k}^{T}+Q_{k+1/k} \end{array}$$
where $S_{k}$ is the covariance of estimated state at time *k* and $Q_{k+1/k}$ is the covariance of the noise in the moving process.

Then, we use the mixture Gaussian model to describe the probability of location at time k+1 for the target:(14)$$\begin{array}{c} P\left(X_{k+1}|Z_{0:k}\right)∼\sum_{j=1}^{n}\mu_{k}^{j}N\left(X_{k+1/k}^{j}\;,\;S_{k+1/k}^{j}\right) \end{array}$$
where $\mu_{k}^{j}$ is the probability of the *j*th motion model at time *k*. Suppose that we obtain *D* measurement points at time k+1, noted as $Z_{k+1}^{d}$, the noise filter process is then to find the measurement that maximizes the probability $P(Z_{k+1}^{d}|Z_{0:k})$ and this measurement is considered the measurement of the target $Z_{k+1}$ at time k+1.

### 2.3. State Estimation and Probability Update

In this step, we follow the Kalman filtering process to calculate the estimated state ${\hat{X}}_{k+1}^{j}$ and the covariance of estimated state $S_{k+1}^{j}$:(15)$$\begin{array}{ccc} G_{k+1} & = & S_{k+1/k}H^{T}{\left(HS_{k+1/k}H^{T}+R_{k+1}\right)}^{-1} \end{array}$$(16)$$\begin{array}{ccc} {\hat{X}}_{k+1}^{j} & = & X_{k+1/k}^{j}+G_{k+1}\left(Z_{k+1}-HX_{k+1/k}^{j}\right) \end{array}$$(17)$$\begin{array}{ccc} S_{k+1}^{j} & = & \left(I-G_{k+1}H\right)S_{k+1/k} \end{array}$$
where $R_{k+1}$ is the covariance of the noise in a measurement and $G_{k+1}$ is the Kalman gain.

After the estimation of the current state, we update the probability of each motion model. The measurement $Z_{k+1}$ can be expressed by the *j*th motion model with the probability:(18)$$\begin{array}{c} P\left(Z_{k+1}|{\hat{X}}_{k+1}^{j}\right)∼N\left({\hat{X}}_{k+1}^{j}\;,\;S_{k+1}^{j}\right) \end{array}$$

As we have the probability of each motion model $\mu_{k}^{j}$, which is the same as the probability of ${\hat{X}}_{k+1}^{j}$, the probability of the *j* motion model at the current time step is obtained using the Bayesian rule:(19)$$\mu_{k+1}^{j}\propto P\left(Z_{k+1}|{\hat{X}}_{k+1}^{j}\right)\mu_{k}^{j}$$

Now, we obtain the estimated state, the measurement, and the updated motion model probability; we then follow the first trajectory prediction process to predict the location of the target at the next time step, by which we can iteratively track and update the state of the target.

### 2.4. Back-Tracking Correction

In a Lidar system, the false-alarm rate $P_{fa}$ is used to describe the ratio of points generated by noise to all measured points in the radar scan. Thanks to the application of TCSPC technology, the false-alarm rate $P_{fa}$ is generally very low (lower than 5%). When the sampling frequency of the Lidar is sufficient, the effect of false-alarm points on target tracking is negligible due to accurate estimation of the target’s state. However, when the radar sampling frequency is insufficient, even a low false-alarm rate can have a large impact on the performance of target tracking due to the large errors in the estimation of the target state during tracking. In the bird target tracking process presented above, there is always a probability that a false-alarm point will be taken as the measurement of a real target during the noise filtering process. The mistaken measurement point therefore interferes with the following L−1 steps of target tracking even though the following measurements are correct because the GRU-based trajectory prediction models require *L* historical data as input. In order to reduce the impact of incorrectly selected false-alarm points on the following target tracking process, we propose a simple back-tracking correction mechanism. The principle of the back-tracking correction mechanism is explained in Figure 5.

At the noise filtering step of time *k*, we can calculate the probability that a measurement $Z_{k}^{d}$ is the measurement of a real bird target $P(Z_{k}^{d}|Z_{0:k-1})$, as shown in Equation (Equation 14). We sort the measurement points by probability from highest to lowest and keep at most the first $D^{\prime }$ points that satisfy the condition:(20)$$\begin{array}{c} P(Z_{k}^{d}|Z_{0:k-1})\leq \eta \end{array}$$
where η is the selection threshold. If none of the measurements satisfy this criteria, we can release the constraint on η to choose at least one measurement point. For the $D^{\prime }$ points, we choose the point with the maximum probability as the measurement to continue the tracking process, as designed previous. The remaining points are kept and called alternative points; we treat each alternative point separately as a real measurement point and continue the target process independently until the noise filtering step at time k+1. Suppose that we have $D^{\prime }$ prediction areas at time k+1 and $D^{\prime }$ new measurement points satisfying the threshold criteria shown in Equation (Equation 20); then, there are $D^{\prime \;2}$ combinations. Then, we will select the measurement point $Z_{k+1}^{d_{j}}$ and the historical trajectory {$Z_{1:k-1},Z_{k}^{d_{i}}$} with the maximum probability $P(Z_{k+1}^{d_{j}}|Z_{1:k-1},Z_{k}^{d_{i}})$ to continue the tracking process. The remaining $D^{\prime }-1$ different combinations with the next highest probability are alternative historical trajectories.

Since the false-alarm rate $P_{fa}$ is very low, the probability that two consecutive false-alarm points are considered measurements of the real target is less than $P_{fa}^{2}$ and can be ignored. We note $Z_{k}^{t}$ as the measurement of the target and note $Z_{k}^{f}$ as the false-alarm point at time *k*. Suppose that we have a mistake at time step *k*, since the measurements of a target have time-related relevance and the false-alarm points appear randomly, we have the following:(21)$$\begin{array}{c} P\left(Z_{k+1}^{t}|Z_{0:k-1},Z_{k}^{t}\right)>P\left(Z_{k+1}^{t}|Z_{0:k-1},Z_{k}^{f}\right) \end{array}$$

As a result, we take the trajectory {$Z_{0:k-1},Z_{k}^{t},Z_{k+1}^{t}$} to continue tracking, and correct the mistake made at time *k* by the back tracking correction mechanism.

In this mechanism, the threshold η ensures that most of the obvious false-alarm points are filtered out, which preliminarily reduces the measurement point quantity. Then, we keep $D^{\prime }$ different historical trajectories at most at each time step, which avoids an exponential increase in complexity and can effectively further reduce the number of possible historical trajectories. The proposed mechanism corrects the mistake that a false-alarm point is considered as a real target and reduces its impact without introducing high complexity. Therefore, we keep at most $D^{\prime }$ trajectories per loop, and at least one trajectory when only one combination satisfies the two criteria. The performance and complexity depend on the choice of $D^{\prime }$, η, the false-alarm rate, and the prediction error in actual operation. We will evaluate the mechanism and explain it later in the Results Analysis section.

## 3. Experimental Setup

### 3.1. Data Preparation

In this research, a pigeon trajectory dataset is used [36]. There are three pigeon groups in the trajectory dataset, and each group contains 10 pigeons. The trajectory is obtained from the GPS device on the pigeons, and the trajectory information contains three-dimensional position, velocity and acceleration data with a sampling interval of 0.1 s. Since pigeons stop flying from time to time, a new trajectory begins when the pigeon starts to fly again. Even though the trajectories of different pigeons belonging to the same flock have a similarity with each others, it can be assumed that each trajectory has a uniqueness due to the splitting and aggregation of pigeon flocks. Therefore, each pigeon has its own unique trajectory, and we track a pigeon on its unique trajectory. In order to simulate Lidar scanning data, only the horizontal position information (X,Y position) are used in the simulation, and the velocity information is obtained by differentiating the position information during the simulation. The original data in the dataset are downsampled at different sampling time intervals Δt from 0.1 s to 5 s with 0.1 s step. Finally, all trajectories are divided into a training set, a validation set, and a test set with a ratio of 8:1:1. In order to simulate real scenario noise on a Lidar system, we add white Gaussian noise with the probability distribution $N(0,\;4\delta l^{2})$, where δl is Lidar detection resolution that depends on the average distance between the target and the Lidar location [37], and thus, δl varies from 0.5 m to 3 m in the simulation.

### 3.2. Implementation Details

In this paper, five GRU-based motion models are implemented for IMM-GRU method. Each motion model has the same architecture: the GRU layer *M* is set to 2, the dimension of hidden features is set as 32, and the input data length *L* is set to 7. In order to learn the characteristics of different motion change information, the downsampled trajectories are divided into five categories according to Δt∈{[0.1,1],[1.1,2],[2.1,3],[3.1,4],[4.1,5]}: the data with Δt∈[0.1,1] and [4.1,5] contain, respectively, the slowest and fastest motion change information. Then, the datasets are used to correspondingly train the five different motion models. The motion models are named from motion model-1 to motion model-5, which motion model-1 designed to learn the slowest motion change information trained using a dataset with Δt∈[0.1,1] and the motion model-5 designed to learn the fastest motion change information. In order to prevent the GRU model from overfitting, we add a dropout layer between the hidden layers, and the dropout probability is set to 0.1. In the IMM-GRU method, the measurement noise covariance R is set as 4I, corresponding roughly to the noise level at the measurement process, and the system noise covariance Q is set as 16I, which reduces the confidential level of the system transition process to enable fast adjustment when target motion changes occur. The input signal of the GRU network at each time step is normalized before feeding into the GRU network. For the back tracking correction mechanism, the maximum historical trajectories $D^{\prime }$ and the threshold η are set, respectively, to 2 and 0.01.

Classical methods, such as IMM-UKF, UKF, PF, and GRU-EKF, are implemented. We set 1000 particles in the PF method in order to obtain the best possible performance. The conventional GRU deep neural network used in the GRU-EKF method consists of eight layers, the dimension of hidden features is set to 128, and the input length is set to 7. The conventional GRU deep neural network is trained with the entire training dataset, with Δt∈[0.1,5]. We use the same dropout method, and the dropout probability is set to 0.1. In the GRU-EKF method, the GRU network gives the prediction, and the EKF method is used to calculate the estimated state as presented in the IMM-GRU method.

All the GRU-based neural networks are trained with the corresponding training dataset and are optimized with the Adam with a 0.5 weight decay for five epochs. The training process is stopped when the loss no longer decreases in the last 20 epochs. The other hyper-parameters are as follows: the batch size is 256, and the learning rate is initially set as 0.01 and is optimized using a cosine annealing schedule, each with five epochs. We choose the mean square error loss function to compute the loss. The models are implemented in PyTorch and trained on a workstation with a GTX 3090Ti GPU.

### 3.3. Experiment Design

For the evaluation of the tracking performance, we choose root mean square error (RMSE) and false detection ratio as the evaluation metrics. The RMSE is calculated as follows:(22)$$\begin{array}{c} RMSE=\sqrt{\frac{1}{N}\sum_{k=1}^{N}{\left(z_{k}-X_{k}\right)}^{2}} \end{array}$$
where *N* is the total time step of the trajectory used in simulation; $X_{k}$ is the real state value at time *k*; and $z_{k}$ will be, respectively, estimated state ${\hat{X}}_{k}$ and prediction $X_{k/k-1}$ for the evaluation of the estimation and prediction performance.

The false detection ratio $P_{f}$ is calculated as follows:(23)$$\begin{array}{c} P_{f}=\frac{N_{f}}{N_{t}} \end{array}$$
where $N_{f}$ is the number of events in which a false-alarm point was treated as a real target measurement during the simulation and $N_{t}$ is the number of false-alarm point occurrences. This metric can be used to evaluate the algorithm’s resistance to false-alarm point interference.

By analysing the mechanism of target tracking, there are several factors that affect the tracking performance, such as the sampling time interval, the false-alarm rate, and the way the motion mode changes, where the way the motion mode changes can be reflected in the different trajectories of the target motion. Moreover, the parameters in the proposed method, such as the number of motion models, the configuration of the motion model, and the input mechanism setting, have also directly impacted the tracking performance and should be evaluated. Therefore, we set up the following experiments to compare the performance of our proposed algorithm with the conventional method to validate the technical contribution previously listed and to study the impact of the parameters in the proposed method:(1)The evaluation of prediction and estimation performance at different sampling time intervals to show that the proposed method can overcome the defect due to a low sampling frequency.(2)The evaluation of the function of motion models.(3)The evaluation of the number of motion model in IMM-GRU.(4)The evaluation of the probability of false-alarm points.(5)The evaluation of the back-tracking correction mechanism.

## 4. Results Analysis

### 4.1. Performance at Various Sampling Time Intervals

The performance of prediction and state estimation is evaluated through RMSE versus sampling time interval Δt. The false-alarm rate is set to 0. The results of the test set shown in Figure 6 are the average from 500 Monte Carlo experiments at different trajectories. The noise added to each trajectory is white Gaussian noise with zeros mean and 4 $m^{2}$ variance.

From the Figure 6a, we can see that the IMM-GRU method gives generally better prediction and estimation than other classical methods, especially when Δt is larger than 2 s, and over 18 m prediction performance gain compared to the sub-optimal solution, which proves that the GRU-based network succeeds in learning the motion change information and gives better prediction. IMM-UKF, UKF, and PF are slightly better than IMM-GRU when the Δt is between 0.1 s and 0.4 s, IMM-GRU has about 0.5 m RMSE more at Δt= 0.1 s. This fact shows that the trajectory between any two observation moments can be approximated as a linear-like motion when the sampling interval is small, which requires less adaptability to the variable motion modes of the tracking method. GRU-EKF has similar performance to that of IMM-GRU due to the stacked learning of five small networks and the interacting motion model selection mechanism; IMM-GRU has a better RMSE performance and a lower computational load. For the estimation performance shown in Figure 6b, it has similar characterization to the prediction performance shown in Figure 6a: the IMM-GRU method has nearly identical performance when the Δt is small and has the best estimation performance when the Δt is large, and the RMSE of the estimation is only about 4.5 m at Δt = 5 s, a 26% improvement compared with the sub-optimal solution GRU-EKF. The results prove that the proposed method can significantly improve the prediction and state estimation performance and can achieve bird tracking at low observation sampling frequencies.

In terms of computational complexity, the average single-step tracking computation times of the IMM-UKF, UKF, PF, GRU-EKF and IMM-GRU methods are 0.74 ms, 0.23 ms, 50.24 ms, 23.71 ms, and 13.65 ms, respectively, all of which are less than 100 ms and satisfy the real-time computational requirements of the program. Since IMM-UKF and UKF are model-based methods, they have less computational complexity than data-based methods, such as the GRU-EKF and IMM-GRU methods. The PF method is the most time-consuming since we have configured 1000 particles in the PF method; even though large search particles have been configured, its performance is much worse than the data-based methods GRU-EKF and IMM-GRU. Compared with the GRU-EKF method, the proposed IMM-GRU reduces the computation time by about 43%, which proves the significant improvement in computational complexity.

### 4.2. Motion Model Function Evaluation

In this subsection, we design simulation tests in order to study the function of each motion model during tracking and to investigate the conditions that are favourable for our method. We first test the proposed method on a circular trajectory. In the circular trajectory, the target makes a turning movement with a constant turning rate, so the target has only one motion. The tracking results at Δt = 1 s are shown in Figure 7. The tracking trajectory is shown in Figure 7a, and the RMSE of the estimation is shown in Figure 7b; we can see that the tracking performance is similar for all tested methods. And, the statistical proportion of motion model weights during tracking is shown in Figure 7c. We can see that motion model-1, which learned the slowest motion change information in IMM-GRU, takes up 100% of the weight. The results show that IMM-GRU recognizes that the target is in a state of slow motion change, which corresponds to the circular trajectory that has only one mode, and the motion change rate is zero.

Then, we increase the sampling time interval from 1 s up to 5 s with a 1 s step for the circular trajectory, the estimation RMSEs of each method are shown in Table 1. We can see that the tracking performance is similar for all tested methods at Δt = 1 s, 2 s, and 3 s. This is because the circular trajectory contains only one simple motion mode, and the increase in the sampling time interval is only manifested in the model as the increase in velocity or angular velocity, and the model is still accurate, so the performance of IMM-GRU is similar to that of conventional algorithms. At Δt = 4 s and 5 s, the inaccuracies of the model introduced by the large sampling time interval is not negligible; therefore, the IMM-GRU begins to show its advantage in tracking a target under a large sampling time interval and the fast motion change condition. The results confirm that the IMM-GRU has advantages at large sampling intervals and has similar tracking performance to that of conventional algorithms when the target is in a state of slow motion change.

Afterwards, we tested the methods on a pigeon trajectory that was randomly selected, and the trajectory was resampled with different sampling time intervals. The results are shown in Figure 8: Figure 8a,d,g show, respectively, the tracking results on the same trajectory under sampling time interval Δt = 1 s, 3 s, and 5 s; Figure 8b,e,h show, respectively, the corresponding tracking errors of each method; and Figure 8c,f,i show, respectively, the statistical results of the motion model weights at any time step during tracking. Since the pigeon changes its flight state during its flight, going through various states such as straight flight, left turn, and right turn, the pigeon trajectory contains different motion states. We can see that the tracking trajectory exhibits complex and variable characteristics, and the tracking error becomes increasingly large with increasing sampling time. When Δt is small (at 1 s) and the motion changes slowly, all the methods under test have similar performance, as shown in Figure 8b. When Δt is large (at 5 s) and the motion changes quickly, the proposed IMM-GRU has not only a minimal average error but also the most stable performance with small variation, as shown in Figure 8h. For Δt = 3 s shown in Figure 8e, IMM-GRU has shown significant estimation performance improvement compared with conventional methods, while the results obtained on the circular trajectory at Δt = 3 s do not show the fact that IMM-GRU has a clear improvement. Since the pigeon trajectory contains multiple motions and more complex motion change information than the information contained in the circular trajectory, we can see that the proposed method has a better performance in the complex situation caused by the changes in the complex motions.

For the motion models, motion models-1, -2, and -3, which were designed to learn slow motion change information, dominate IMM-GRU at Δt = 1 s. Compared with the results on the circular trajectory in which only one motion model took 100% of the weight, shown in Figure 7c, we can clearly see that there are multiple motions in pigeon trajectories, and it is not possible to describe all the motion changes with only one motion model. In contrast, at Δt = 5 s, motion models-4 and -5, which were designed to learn fast motion change information, dominate the weight. And, motion models-1 and -2 are still useful because the motion of the target over a number of time periods can be equated to a slowly changing motion. The results prove that all the motion models implemented in the IMM-GRU are useful at large sampling time intervals, while only the motion models designed for slow motion change information are useful at a small sampling time interval. We can conclude that, with the sampling interval, each motion model can contribute differently and specifically to the tracking of a manoeuvring target. With the benefit of this innovation, IMM-GRU can reach a better performance in tracking, especially for fast changing motions and/or large sampling time interval.

### 4.3. Evaluation of Motion Model Quantity

We explored the IMM-GRU prediction performance by varying the number of motion models with the same GRU configuration. The training dataset is divided into the corresponding number of parts, and each GRU-based motion model is trained to convergence in the same way as the previous configuration. The results of the RMSE performance over different sampling time intervals are shown in Figure 9. We can see that the performance degrades significantly when the number of model decreases to 3 compared with the performance with five motion models, since a small network cannot learn all information from the corresponding training dataset. In contrast, while the number of model increases, the prediction performance gains little. The results prove that five motion models are a good compromise between performance and complexity.

### 4.4. Evaluation over False-Alarm Rate

In a Lidar operation, measurement data from Lidar scanning contains actual target measurements as well as false-alarm points due to background light noise. Due to the application of photon-counting technology, the false-alarm rate of Lidar is low, generally lower than 5%. In this subsection, we compare the different methods at 2% and 5% false-alarm rates to verify the performance of the algorithms against background light noise interference. The experimental results are shown in Table 2, which lists the estimation RMSE of each method under different false-alarm conditions, and the data in the parentheses are the changes in tracking errors relative to the tracking errors under the condition of no false-alarm points. It can be seen that the tracking performance of IMM-GRU is the best under different false-alarm rates; especially under the condition of large sampling intervals, the tracking error of IMM-GRU method still has obvious advantages over the other methods. The experiment results prove that the IMM-GRU method has good false-alarm resistance.

### 4.5. Evaluation of Back Tracking Correction Mechanism

In this subsection, we will evaluate the function of the back-tracking correction mechanism. We perform IMM-GRU with and without the back-tracking correction mechanism on different trajectories, and we add a false-alarm point with 5% probability to the trajectories. Then, the RMSE of the state estimation and the false detection ratio of each method are calculated and listed in the Table 3. From Table 3, we can see that IMM-GRU with the back-tracking correction mechanism has a better performance and a lower false detection ratio; the back-tracking correction mechanism can correct the mistake that treats a false-alarm point as a measurement of a real target in the previous step, thus attenuating the interference caused by the mistake. The results prove the effectiveness of the back-tracking correction mechanism.

In the back-tracking correction mechanism, the parameter $D^{\prime }$ is used to configure the maximum historical trajectories kept at each step. It is obvious that the large $D^{\prime }$ means that more trajectories are kept as alternatives, which leads to a high computational load, and the back-tracking correction mechanism is closed when $D^{\prime }$ = 1. We evaluate the back-tracking correction with $D^{\prime }$ set to 2 and 3 at Δt = 3 s. The RMSE of the estimation is 3.08 m at $D^{\prime }$ = 2 and 3.05 m at $D^{\prime }$ = 3. The results show that a larger $D^{\prime }$ does not gain much in terms of RMSE performance because the false-alarm rate is low and thus the extra alternative is redundant. In addition to the $D^{\prime }$, the η also affects the performance and computational complexity of the back-tracking correction mechanism. The large η will result in only one measurement being selected at each time step and thus the back-tracking correction mechanism will in fact not be used, while the small η will result in lots of measurement points and thus high computational complexity. Moreover, the setting of η is also directly related to the prediction error: the smaller the prediction error, the larger an η can be set to reduce as many obvious false-alarm points as possible. Under the low-sampling-frequency condition, the high prediction error leads to smaller η settings. The RMSE performance of the estimation at Δt = 3 s with different η configurations are shown in Table 4. The results prove that a lower η can improve the tracking performance but not indefinitely. Therefore, by considering the computational complexity, we set $D^{\prime }$ and η to 2 and 0.01, respectively.

Briefly, we summarize the five different kinds of experiments carried out: The first experiment demonstrated that the proposed method outperforms the IMM-UKF, UKF, PF, and GRU-EKF methods, with at least 26% tracking performance gain at Δt = 5 s. Our method uses the high-sampling-frequency data only in the training process and performs tracking under the low-sampling-frequency condition without any help from the high-sampling-frequency data, which is different from similar work that exists [33,34,35]. For the computational complexity, IMM-GRU meets real-time requirements and reduces the 43% computation time compared to GRU-EKF, which is also a data-driven method. The second experiment shows that the GRU-based motion models indeed learned the motion change information as designed, which is the key point that guarantees the good performance at large sampling time intervals and dynamic manoeuvring target tracking scenarios. The third experiment justifies the configuration of the five different motion models used in IMM-GRU. The fourth experiment demonstrates the good resistance to the false-alarm rate. The fifth experiment shows the further improvement in resistance to the false-alarm rate brought about by the back-tracking correction mechanism. Our research shows that tracking motion change can help to improve the tracking performance of manoeuvring targets, and multiple small GRU networks with the help of the IMM mechanism can achieve better performance than a large complex neural network with less computational complexity.

## 5. Conclusions

In this paper, we proposed an IMM-GRU method for bird tracking on a Lidar system with a low observation refresh rate. The IMM-GRU method leverages multiple GRU networks to match target motions and to predict target trajectories and uses the IMM mechanism to combine different predictions and to give a final appropriate prediction. Simulations on a bird trajectory dataset demonstrate that the IMM-GRU method outperforms other methods such as IMM-UKF, UKF, PF, and GRU-EKF methods. The proposed method shows the good ability of GRU in predicting and tracking a target’s motion change based on temporal information dependencies without any prior model knowledge, which suggests that motion change is effective information in target tracking, and data-driven algorithms can be considered as promising alternatives to model-driven approaches under low-sampling-frequency conditions. The IMM mechanism allows for using several small GRU networks in parallel and thus reducing the computational complexity and improving the algorithm performance. The combination strategy can work equally well for other scenarios with diverse model variations. These results highlight the effectiveness of the IMM-GRU method in addressing the tracking challenges posed by low observation rates in Lidar systems. Future work could include the interpretability of the GRU network on a motion model and real implementations of the proposed method on dedicated FPGA-based accelerators for Lidar systems.

## Figures and Tables

**Figure 1 sensors-23-07933-f001:**
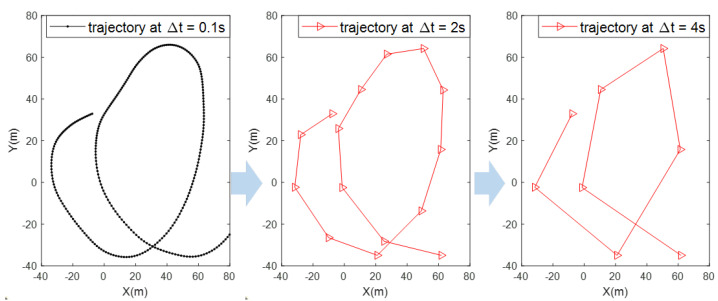
Trajectory at different sampling time intervals.

**Figure 2 sensors-23-07933-f002:**
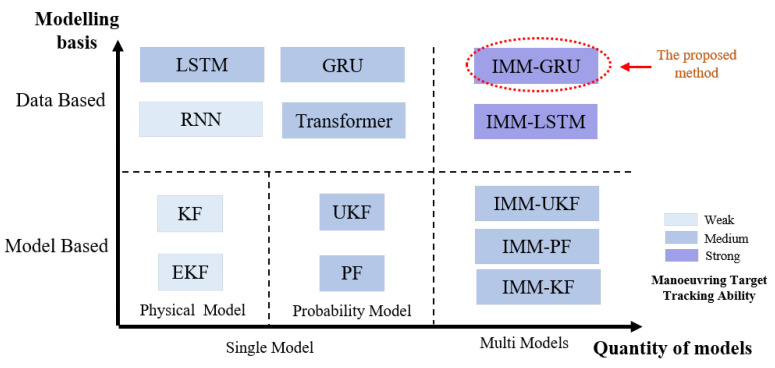
Position of the proposed method in the state of the art.

**Figure 3 sensors-23-07933-f003:**
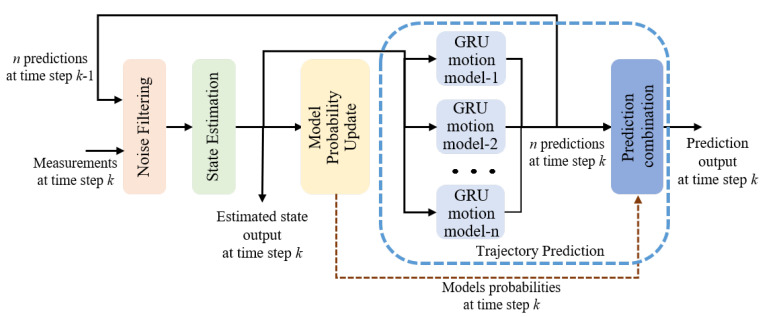
The principle of IMM-GRU.

**Figure 4 sensors-23-07933-f004:**
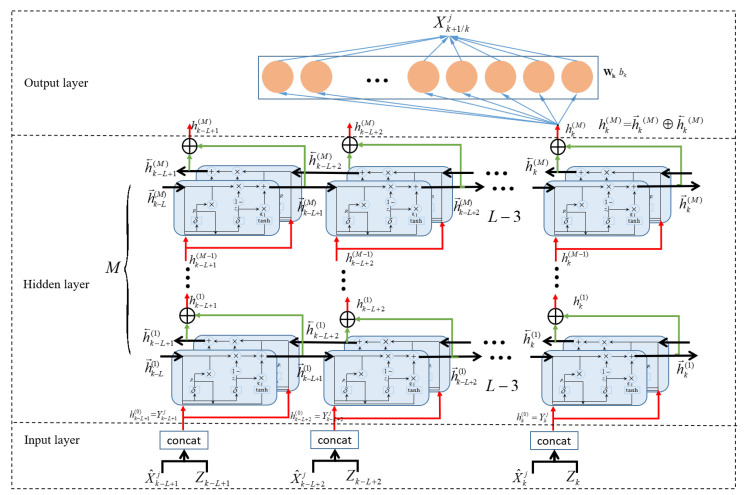
The structure of the GRU-based motion model.

**Figure 5 sensors-23-07933-f005:**
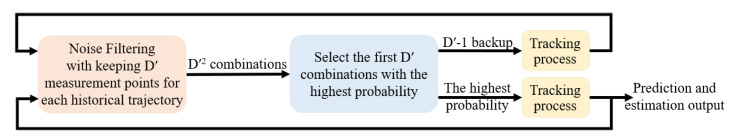
The principle of back-tracking correction.

**Figure 6 sensors-23-07933-f006:**
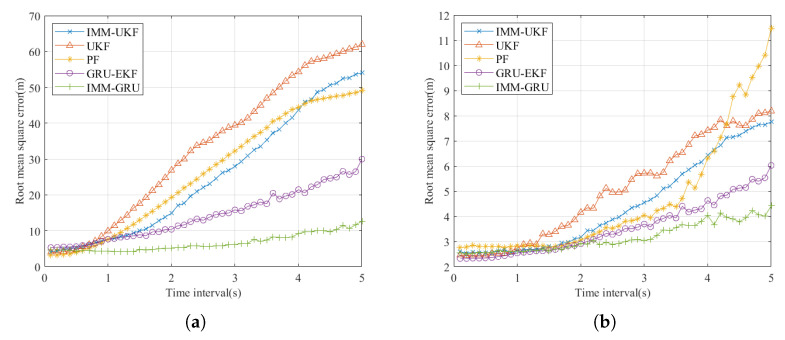
Comparison of (**a**) prediction performance and (**b**) estimation performance at varying sampling time intervals.

**Figure 7 sensors-23-07933-f007:**
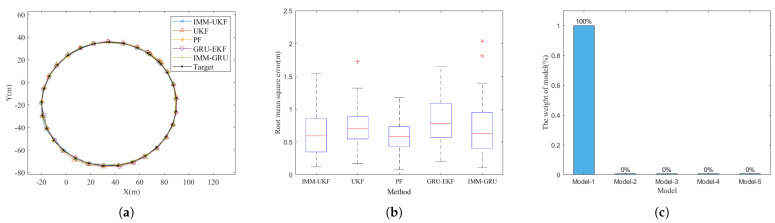
Tracking target on a circular trajectory. (**a**) Tracking trajectory at Δt = 1 s. (**b**) RMSE performance at Δt = 1 s. (**c**) The statistic proportion of motion model weights in tracking at Δt = 1 s.

**Figure 8 sensors-23-07933-f008:**
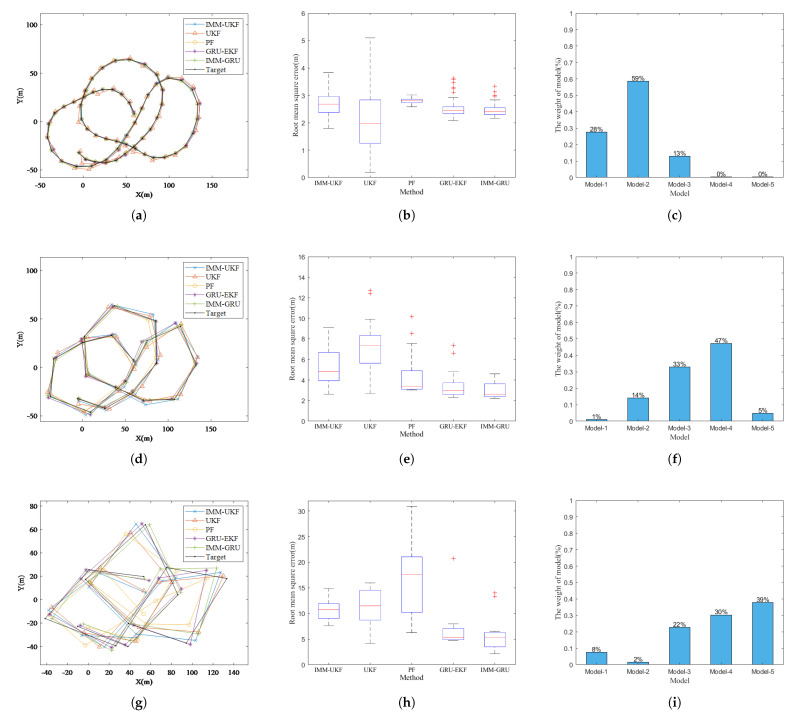
Tracking on a same trajectory with different sampling intervals. (**a**) Tracking trajectory at Δt = 1 s. (**b**) RMSE performance at Δt = 1 s. (**c**) The statistic proportion of motion model weights in tracking at Δt = 1 s. (**d**) Tracking trajectory at Δt = 3 s. (**e**) RMSE performance at Δt = 3 s. (**f**) The statistic proportion of motion model weights in tracking at Δt = 3 s. (**g**) Tracking trajectory at Δt = 5 s. (**h**) RMSE performance at Δt = 5 s. (**i**) The statistic proportion of motion model weights in tracking at Δt = 5 s.

**Figure 9 sensors-23-07933-f009:**
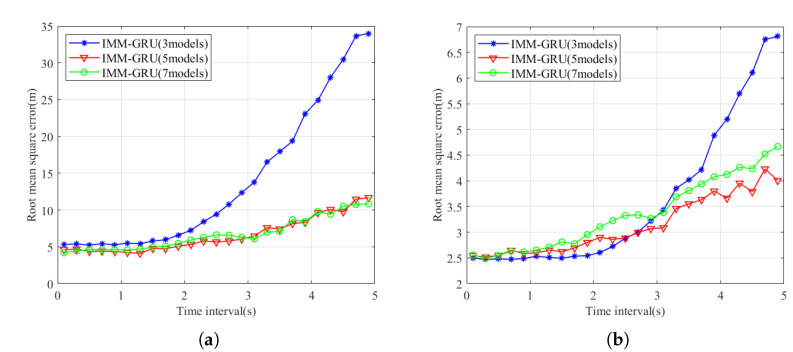
Comparison of prediction and estimation performance at varying numbers of motion model. (**a**) Prediction performance. (**b**) Estimation performance.

**Table 1 sensors-23-07933-t001:** Tracking target on a circular trajectory at different Δt.

Δt (s)	IMM-UKF (m)	UKF (m)	PF (m)	GRU-EKF (m)	IMM-GRU (m)
1	2.48	2.54	2.80	2.46	2.46
2	2.52	2.66	2.81	2.84	2.72
3	3.15	3.45	2.89	3.49	2.87
4	4.88	6.03	3.92	3.46	3.53
5	7.20	8.71	8.25	4.03	4.44

**Table 2 sensors-23-07933-t002:** Tracking performance at different false-alarm rates.

False-Alarm Rate	Δt (s)	IMM-UKF (m)	UKF (m)	PF (m)	GRU-EKF (m)	IMM-GRU (m)
2%	1	2.63 (+0.33)	3.07 (+0.01)	2.73 (+0.08)	2.54 (+0.25)	2.57 (+0.02)
	2	3.21 (+2.04)	6.20 (+0.05)	3.13 (+0.08)	3.11 (+0.45)	2.95 (+0.03)
	3	4.80 (+3.14)	8.86 (+0.26)	3.99 (+0.05)	3.85 (+0.40)	3.06 (+0.04)
	4	6.91 (+4.13)	11.55 (+0.48)	6.68 (+0.36)	5.38 (+1.01)	4.07 (+0.12)
	5	8.33 (+4.47)	12.65 (+0.57)	11.64 (+0.15)	6.30 (+0.53)	4.86 (+0.52)
5%	1	2.64 (+0.38)	3.11 (+0.00)	2.82 (+0.01)	2.55 (+0.26)	2.59 (+0.04)
	2	3.37 (+2.40)	6.56 (+0.20)	3.11 (+0.04)	3.20 (+0.54)	2.99 (+0.07)
	3	5.34 (+3.78)	9.49 (+0.80)	4.08 (+0.06)	4.02 (+0.58)	3.08 (+0.06)
	4	7.79 (+4.99)	12.40 (+1.36)	7.26 (+0.94)	5.67 (+1.29)	4.38 (+0.43)
	5	8.72 (+5.36)	13.54 (+0.96)	12.32 (+0.84)	7.17 (+1.40)	4.87 (+0.53)

**Table 3 sensors-23-07933-t003:** Tracking performance between with/without back-tracking correction mechanism.

Δt (s)	IMM-GRU with Back- Tracking Correction Mechanism		IMM-GRU without Back- Tracking Correction Mechanism	
	Estimation RMSE (m)	False detection ratio	Estimation RMSE (m)	False detection ratio
1	2.59	0.6%	2.70	0.8%
2	2.99	1.0%	3.11	1.2%
3	3.08	1.6%	3.21	2.2%
4	4.38	3.4%	4.62	4.2%
5	4.87	4.4%	5.10	5.6%

**Table 4 sensors-23-07933-t004:** The RMSE performance of the estimation at Δt = 3 s with different η configurations.

η	0.005	0.01	0.02
RMSE (m)	3.07	3.08	3.15

## Data Availability

The data presented in this study are available from the corresponding author upon request. The data are not publicly available due to privacy reasons.

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
