# Peer review of "A Gated-Recurrent-Unit-Based Interacting Multiple Model Method for Small Bird Tracking on Lidar System"

_sensors, 2023, doi:10.3390/s23187933_

Round 1
Reviewer 1 Report
This paper proposes an IMM-GRU method for bird tracking on Lidar system with low observation refresh rate, and this method combines IMM and GRU to solve problems appearing in bird target tracking: the motion model cannot be represented at low sampling frequency; false alarm point filtering. The simulation results also show good performance, but some problems still need to be corrected:
1. The word “maneuvering” is wrong and inconsistent before and after. Such as P1 using the “maneuvering”, while the line 87 for P3 using “manoeuvring”. The expression of the “neural network” has the same problem, such as the 325 line for P9 using “neutral network”, while “neural network” used in other positions, and similar problems can be found in other positions of the paper.
2. Some related work can be properly supplemented between the introduction and the proposed method. Such as methods adapted to low sampling frequency, filter selection, etc.
3. The forward hidden feature of P6 uses the same symbol with the backward hidden feature, thus the expression is wrong.
4. The 496 line for P15 about the selection of value
can use the form of chart or table to more intuitively express the rationality of parameter selection.
5. The conclusion indicates that IMM-GRU method outperforms other methods, but the other methods listed need to correspond to the methods in the simulation experiment, such as IMM-GRU is not compared with GRU.
Moderate editing of English language required
Reviewer 2 Report
Dear Authors.
First, I would like to thank the authors for their work and determination in carrying out this study. The manuscript entitled "A Gated Recurrent Unit Based Interactive Multiple Model Method for Small Bird Tracking on Lidar System" contains:
- Introduction to the manuscript.
- IMM-GRU Method
- Experimental Setup
- Results Analysis
- Conclusions.
The manuscript presents the case study, which shows the results of the proposed IMM-GRU method for tracking birds in the airport environment on a Lidar system with a low refresh rate of observation. In the simulation, the authors used a circular trajectory, and in the experiment, the authors used three groups of pigeons with a GPS device, each containing ten pigeons.
I have comments on the following manuscript:
Line 188 - different variables are marked the same - forward hidden feature and backward hidden feature.
Line 290 – 30 pigeons were used in the experiment, and line 422 – Fig. 7a, d, g shows the trajectory of the pigeon with a tracking interval of 1, 3, and 5 seconds. Based on what rule are the results of tracking the flight of one pigeon with a GPS device displayed? How did the trajectories of the remaining 29 pigeons perform in terms of prediction performance and estimation performance?
Discussions - I need to include a comparative debate about the results achieved with previous scientific works.
The manuscript has other formal errors or deficiencies, which will be removed during the possible process of checking the English and final editing of the contribution by the editors according to the valid standards of the journal.
Best regards
Reviewer 3 Report
The paper is introduced a new method-based Interactive Multiple Model (IMM) approach for tracking bird targets at low sampling frequencies. The authors note that the proposed method outperforms classical tracking methods at low refresh rates with at least a 26% improvement. The paper contains significant theoretical results and new and significant experimental results.
However, it could be improved:
1. The title, abstract, and introduction were found appropriate. It is positive that the main contribution and the general organization of the article are presented, especially in the introduction. The author analyzed literary sources. But minor comments include describing the analysis of more contemporary literary sources. For example,
- DOI:10.1098/rsif.2007.1349;
- https://ceur-ws.org/Vol-3426/paper14.pdf.
It is not a must to include those above, but references to related works have to be improved and updated.
2. I recommend the authors put a chart in the paper to illustrate the presented algorithms. This will help the reader to understand the paper easily. At least qualitative comparisons with previous research works should be included in the paper.
3. The authors present the average time of one-step tracking calculations and note that the results obtained meet the program's requirements for real-time computing applications. It would be more expedient to investigate and provide indicators of the speedup obtained in comparison with classical modern methods.
4. The authors should also improve the quality and readability of the figures. In particular, Figures 5-8.
5. Overall, the conclusion is well structured, but it could be improved by providing more details and clarity on the specific experiments that were conducted and the results obtained, as well as the main findings and contributions of the paper. And also describe the prospects for future research.
Reviewer 4 Report
1) In the title: “Interactive method” is opposite to “automatic method”. Do you mean a user interaction here? Observe, that the original paper is using the term “the interacting multiple model algorithm” which can be interpreted as an (automatic) interaction between different models of an algorithm. I suggest to use the term “ many interacting models”.
2) In the Abstract: “ … outperforms classical tracking methods at low refresh rates with at least 26% improvement.”
This statement is not precise. The improvement of which metrics is observed?
3) Figure 2 contains is extremely unclear and, when given the later following explanation, it contains severe problems: 1. The system has no input (an initial measurement or an initial state estimation?), 2) Data streams are not clearly separated from control signals, 3) No discrete time index is provided – a continuous-time feedback? 4) The model probability block should control the “Prediction combination” and not the input to the GRU networks (see eq. (8)) ? 5) The feedback from “Prediction combination” to “Noise filtering” is unclear (there are n predictions with model probabilities send back).
4) The input block in figure 3 is unclear. What is the meaning of “Xhat_k | Z_k”? The symbol “Xhat_k” means an estimate of state X at time k which is obtained based on Z_{1:k}. In the text it is suggested that “X_hat | Z “ means concatenation. Use a proper symbol for this operation. In equation (3) this concatenated input to layer 1 is denoted as Y but Y is denoted as output of every layer.
The output layer is also misleading – there are no weights W visible, while the hidden neurons are denoted as Y_k, where Y_k is also denoted as the input vector to this FC layer.
5) In the subsection 3.1 “Data preparation” an important information is missing about the “training-to-testing” split and the number of data samples used for training and testing.
6) In section 4 “Results analysis” it is unclear whether the results represent training- or test performance.
Remarks:
1) Line 60: “estimate state” à estimated state
2) Line 68: the original title of ref. [14] is “Interacting … algorithm” and not “Interaction …”.
3) Line 71: ”authors combines” à … combine
4) Line 101: explain the term “false alarm point filtering”. This problem has not been introduced so far.
5) Line 159: “all the process are” ? à all processes
Round 2
Reviewer 1 Report
The authors have replied my comments and I suggest that the journal accepts this manuscript in present form.
Minor editing of English language required